# Contribution of Sentinel-3A Radar Altimetry Data to the Study of the Water Level Variations in Lake Buyo (West of Côte d'Ivoire)

Sékouba Oularé [1,2,*], Valère-Carin Jofack Sokeng [3], Koffi Fernand Kouamé [1,3], Christian Armel Kouassi Komenan [1], Jean Homian Danumah [2], Benoit Mertens [4], You Lucette Akpa [2], Thibault Catry [4] and Benjamin Pillot [4]

[1] Laboratoire des Sciences du Sol de l'Eau et des Géomatériaux (LSSEG), Unité de Formation et de Recherche des Sciences de la Terre et des Ressources Minières (UFR-STRM), Université Félix Houphouët Boigny, Abidjan 22 BP 582, Côte d'Ivoire

[2] Centre Universitaire de Recherche et d'Application en Télédétection (CURAT), UFR-STRM, Université Félix Houphouët Boigny, Abidjan 22 BP 801, Côte d'Ivoire

[3] Unité de Recherche et d'Expertise Numérique (UREN), Université Virtuelle de Côte d'Ivoire, Abidjan 28 BP 536, Côte d'Ivoire

[4] ESPACE-DEV, Univ Montpellier, IRD, Univ Antilles, Univ Guyane, Univ Réunion, 13002 Montpellier, France

[*] Correspondence: sekou.oulare@curat-edu.org or oulare.sekouba@univ-fhb.edu.ci; Tel.: +225-070-772-8850

**Abstract:** The artificial Lake Buyo is an important water reservoir that ensures the availability of water for multiple purposes: drinking water supply, fishing, and energy. In the last five years, this lake has experienced extreme variations in its surface area and water levels, including very significant declines, which has impacted the supply of electricity. This study aimed to assess temporal variations in the water levels of Lake Buyo using radar altimetry. Altimetric data from the Sentinel-3A satellite on Lake Buyo (tracks 16 (orbit 8) and 743 (orbit 372)) were selected over the period from 31 May 2016 to 12 June 2021 and compared to the in situ measurements provided by the Direction de la Production de l'Electricité de Côte d'Ivoire (DPE-CI). The extraction of the time series of the Sentinel-3A altimetric water levels and their corrections (geophysical and environmental corrections) were carried out with the ALTiS software. The results showed an overall agreement between the altimetric water levels and the in situ measurements, with a correlation coefficient ($R^2$) ranging from 0.98 to 0.99 obtained, as well as a Nash–Sutcliffe Efficiency (NSE) coefficient also between 0.98 and 0.99. Further, the bias (0.12 m and 0.13 m) and root mean square error (RMSE) (0.38 and 0.67 m) values showed that the results were acceptable. The analysis of the water levels time series allowed for the identification of two main periods: March to October and November to February. The first period corresponded to a high level period, recording a maximum level of 200.06 m. The second period, from November to March, was characterized by a drop in the water level, recording a minimum level of 187.42 m. The water levels time series provided by Sentinel-3 allowed us to appreciate the respective influences of seasonal and interannual variations on rainfall and the contributions of the Sassandra River tributaries to the water levels of Lake Buyo.

**Keywords:** Sentinel-3A; time series; seasonal variations; Lake Buyo; Côte d'Ivoire

## 1. Introduction

In recent years, decreases in surface water (lakes, rivers) availability for daily use has forced the international community and the United Nations to become involved in the rational management of this issue, in order to achieve the Sustainable Development Goals (SDG) devoted to water (SDG, Chapter 6) and climate (SDG13, Chapter 13). In Côte d'Ivoire, several actions have been carried out to achieve the sustainable development goals, in particular the establishment of the National Environmental Action Plan (PNAE) in 1995, and the National Development Plan (PND) 2016–2020 and 2021–2025. The sixth chapter of the PND 2021–2025 is devoted to "the environment and living conditions", which includes

potable water and sanitation. For the implementation of these actions, good knowledge and monitoring of the water cycle and climate at the regional and local scales, especially that of freshwater (groundwater, lakes, rivers, etc.) is necessary and fundamental to meet the current challenges and reduce uncertainty regarding future changes in water availability.

This study was initiated in this context and focused on Lake Buyo. Lake Buyo is an artificial lake whose water level variation depends on the inflow of the Sassandra and N'zo rivers, and on rainfall. These two rivers are the main tributaries of the Sassandra watershed. The maximum flooded area in the high water level period is 900 km$^2$, while the maximum area of the lake in the low water level period is 230 km$^2$. This lake is an important water reserve that ensures the availability of drinking water and fishing resources for the entire region, which has 103,217 inhabitants. The irrigation of crops, some forms of tourism and sports activities also depend on the lake. Further, a hydroelectric dam was built on the lake to ensure the energy needs of the region but also of a large part of Côte d'Ivoire. The hydrological monitoring of this important lake is necessary in the current context of climate variability.

Traditionally, the monitoring of surface water variability has relied on in situ observations that quantify the movement (height, extent, flow) and quality of water in river channels, lakes, and wetlands [1]. However, in situ networks are scarce and unevenly distributed across the country, especially in isolated areas which are difficult to access or with security concerns. In situ observation and measurement networks are generally expensive to maintain, and the availability of ground-based hydrological stations has decreased significantly in recent decades. Moreover, when they exist, these measurements are most often difficult to obtain due to their heterogeneity in both space and time [2–4]. Today, the European Copernicus program provides data freely to users from satellites such as Sentinel, ENVISAT, etc. These data, which can cover a very long period, are useful for monitoring climate change and its impact on ocean and continental water levels.

Due to the development of earth observation sciences since the 1960s, and especially the development of remote sensing, some natural or anthropogenic phenomena can be monitored in near real time in different places on the planet. Among these phenomena are wetlands' surfaces, oceans and seas, lakes, large rivers, etc. [5–9]. With the advent of some remote sensing techniques, such as radar altimetry, since the 1980s [10–12], it has become possible to monitor variations in oceans [13–16], coastal regions [17–20] and continental water levels (lakes, rivers) [21–26]. Radar altimetry, which was initially used in the oceans, is now applied to continental waters thanks to the diversification of satellites (Topex/Poseidon, ERS 1 and 2, Jason-1/2/3, ENVISAT, SARAL, Sentinel-3A and 3B, Sentinel-6, GFO, Cryosat-2, etc.) and sensors (Poseidon-2,3 and 4, Radar Altimeter (RA), AltiKa, SAR Radar Altimete (SRAL), etc.) [22,27–32]. Satellite radar altimetry covers almost the entire land surface and thus permits researchers to complete, reconstruct and densify the existing in situ reservoir water level datasets to better inform decision making.

The general objective of this work was to analyze the seasonal and interannual variability of the water levels of Lake Buyo using Sentinel-3 radar altimetry data. A detailed description of Lake Buyo is presented in the following section.

## 2. Study Area

Lake Buyo is located between the longitude 06°50′ and 07°22′ west, and the latitude 06°18′ and 06°50′ north in the west of Côte d'Ivoire (West Africa), covering an area of 920 km$^2$ (Figure 1). The hydrological regime of the lake depends on the Sassandra River and the rainfall of the region. The climate is characterized by a transitional equatorial rainfall regime with four seasons, including a long rainy season from March to July and a short rainy season from September to October [33]. The other two seasons are two dry periods from November to March and July to September. The sum of the annual rainfall in Buyo is 942 mm. The annual temperature in Buyo varies between 25 °C and 28 °C, and has an average of 26.25 °C.

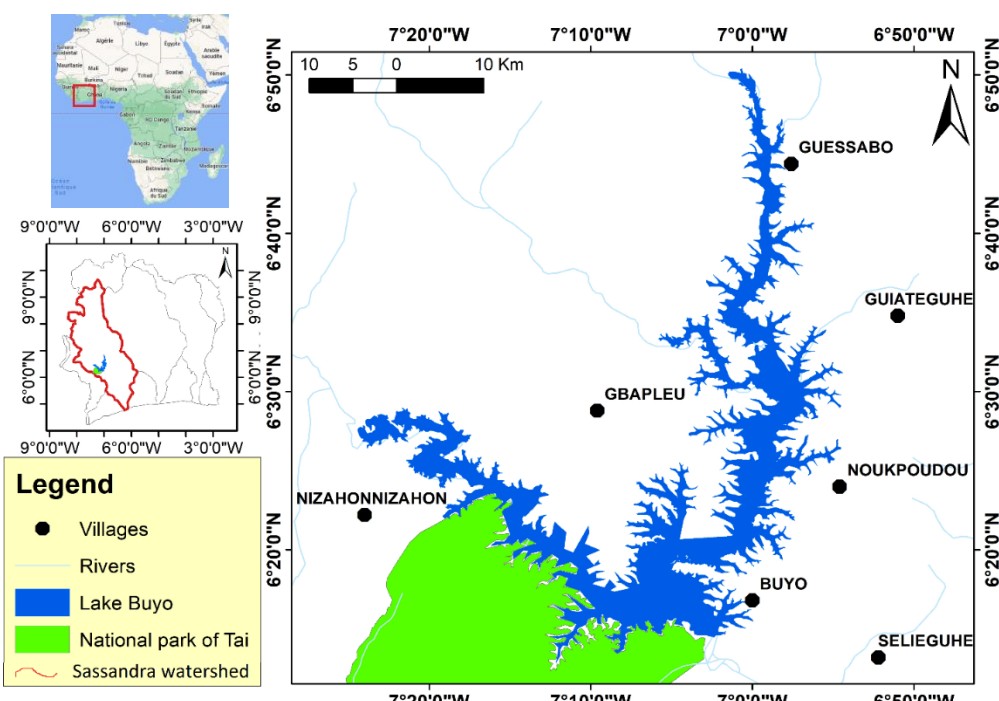

**Figure 1.** Location of Lake Buyo in Côte d'Ivoire (in West Africa).

Lake Buyo is surrounded by semi-deciduous vegetation [34]. A dense forest zone (Taï National Park) can be distinguished in this vegetation formation. The total population of the sites bordering Lake Buyo, according to the data of the General Census of Population and Housing (RGPH), is 103,217 inhabitants [35]. On the socio-economic level, fishing is the second most practiced activity in the locality after agriculture, which is the main economic activity of the active population. Lake Buyo also features important infrastructure for electricity production. The dam built on Lake Buyo produces 25% of the electricity used in Côte d'Ivoire, the second most after that of Soubré, which produces 45% of the electricity supply. In the context of climate variability and change, and considering the outdated hydrological monitoring networks, radar altimetry is an innovative tool that can be coupled with in situ hydrometric measurements to improve the coverage of observed areas through "virtual stations" to ensure regular hydrological monitoring.

## 3. Materials and Methods

### 3.1. Data and Materials

The data used in this work are essentially Sentinel-3A radar altimetry measurements. They were downloaded from the website http://ctoh.legos.obs-mip.fr/ (accessed on 2 February 2021) of the Center for Ocean and Hydrosphere Topography (CTOH). They consist of 56 measurements or cycles for track 16 (orbit 8) and 60 measurements or cycles for track 743 (orbit 372), the duration of a Sentinel-3A cycle being 27 days. The totality of these measurements extends from 2 June 2016 to 12 June 2021 for track 16 and from 31 May 2016 to 10 June 2021 for track 743. These measurements were collected at two virtual stations that correspond, respectively, to the intersections of Sentinel-3A tracks 16 and 743 with the water body (Figure 2).

The in situ hydrometric measurements used for the validation of Sentinel-3A altimeter data were provided by the Direction de la Production de l'Electricité de Côte d'Ivoire (DPE-CI). This department records the daily water levels of all the lakes of Côte d'Ivoire that feature hydroelectric dams. The daily hydrometric measurements collected for this study covered the period from 1 January 2016 to 30 June 2021.

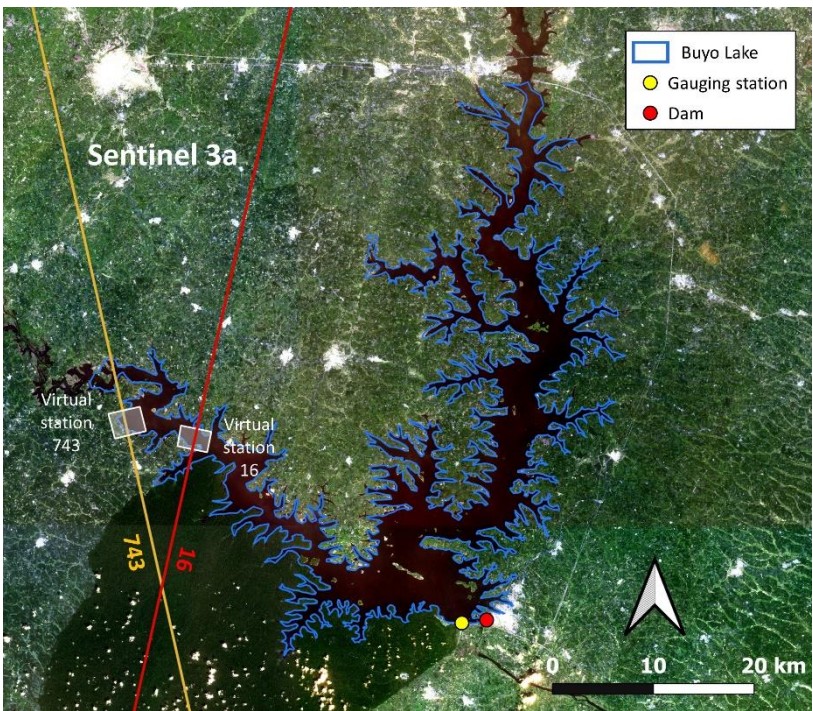

**Figure 2.** Sentinel-3A tracks 16 (orbit 8) and 743 (orbit 372) crossing Lake Buyo.

The main piece of software used for this work was ALTiS (http://ctoh.legos.obs-mip.fr/applications/land_surfaces/softwares/altis (accessed on 2 February 2021)). This software was used to visualize and process radar altimetry data from Sentinel-3A. It has been developed by the Center for Ocean and Hydrosphere Topography (CTOH). It is free, open source software, published under the CeCill license. The second piece of software used in this work was Google Earth. It was used to delineate the intersection of the Sentinel-3A ground tracks with the water body (virtual stations).

*3.2. Methods*

3.2.1. Altimetry Principle

Sentinel-3A altimetry data provide a large number of parameters, some of which can be used to compute water levels of oceans, seas and continental waters. The parameters that are used to calculate water levels are mainly the altitude of the satellite relative to a reference ellipsoid (H), and the range of the satellite, called Range (R) which is the distance between the satellite and the earth's surface. The sum of these two parameters allows us to calculate the water level h according to the following relationship [36–38]:

$$h = H - R \tag{1}$$

where H is the altitude of the satellite relative to a reference ellipsoid and R is the range. Equation (1) is in fact a simplified representation of the measurement of the water level. Indeed, the altitude H and the range R of the satellite contain errors that must be corrected to get closer to the real water level. The corrections applied to these two parameters have been the focus of many studies in the literature [36–38]. Taking into account the different corrections we have the following relationship:

$$h = H - \left( R_0 + \sum \Delta R_{geophyscal} + \sum \Delta R_{environmental} \right) \tag{2}$$

where H is the altitude of the satellite in relation to a reference ellipsoid; this is estimated today with centimeter accuracy using precise orbit determination techniques; $R_0$ is the nadir altimeter distance between the satellite center and the surface of the sea or continental

waters; it is derived from the round trip travel time of the electromagnetic wave emitted by the sensor ($\Delta t$) considering a velocity equal to the velocity of the light in vacuum (C) [38,39]:

$$R_0 = c\Delta t/2 \tag{3}$$

$\sum \Delta R_{geophysical}$ and $\sum \Delta R_{environmental}$ are the sum of the corrections applied to the range to account for atmospheric propagation delays and geophysical effects.

The main geophysical corrections are as follows:

$$\sum \Delta R_{geophysical} = \Delta R_{ion} + \Delta R_{dry} + \Delta R_{wet} \tag{4}$$

where $\Delta R_{ion}$ is the atmospheric refraction distance delay due to the free electron content associated with the dielectric properties of the ionosphere; $\Delta R_{dry}$ is the atmospheric refraction distance correction due to the dry gas component of the troposphere; $\Delta R_{wet}$ is the atmospheric refraction distance correction due to the water vapor and liquid water content of the tropospheric clouds.

For environmental corrections, the following equation is used:

$$\sum \Delta R_{environmental} = \Delta R_{solid\ Earth} + \Delta R_{pole} \tag{5}$$

where $\Delta R_{solid\ Earth}$ and $\Delta R_{pole}$ are the corrections that represent the vertical movements of the earth's crust due to solid earth and polar tides, respectively.

The level-2 GDR (Geophysical Data Records) products provided by CTOH (http://ctoh.legos.obs-mip.fr/applications/land_surfaces (accessed on 2 February 2021)) include all the above geophysical and environmental corrections. These data are relevant for hydrology monitoring [39].

### 3.2.2. Water Level Time Series Extraction

Before extracting water level time series, ALTiS allows us to visualize the hydrological parameters contained in the level-2 GDR, including H, $R_0$ and the different corrections applied to $R_0$ and h. The parameters $R_0$ and h are automatically computed from the Equations (2)–(5) when the data is read, as well as several other variables, such as backscatter coefficients, brightness temperatures, etc. [39].

The first step in the determination of water level time series was the delineation of virtual stations that correspond to the crossing section between Sentinel-3A tracks and the lake. This delineation was carried out using Google Earth. As seen in Figure 3A, track 743 (orbit 372) was going through a major piece of land; only a small part of it was inside the lake. The first virtual station (virtual station 743 (orbit 372)) was therefore defined on this small part of track 743, which was really inside the lake (Figure 3A). The second virtual station (virtual station 16 (orbit 8)) was defined in the same way by identifying the largest section of track 16 (orbit 8) that was inside the lake (Figure 3A). This step eliminated all points located outside the stream, such as floodplains and pieces of land. and selected only valid points that were located inside the stream. The second step consisted of exporting these virtual stations into the Altis software in order to calculate the water level time series. In this step, a second verification was performed to manually remove aberrant points that may have appeared in the virtual stations by analyzing their elevations (Figure 3B). The computation of the altimetry water levels of the lake can then be performed either with the average or with the median of all the points actually located inside the lake; in this study, the median was used. Water level time series calculated with the median are generally better than those calculated with the mean values [39–41]. This process was repeated for each cycle and permitted us to build water level time series at the virtual stations (Figure 3C,D).

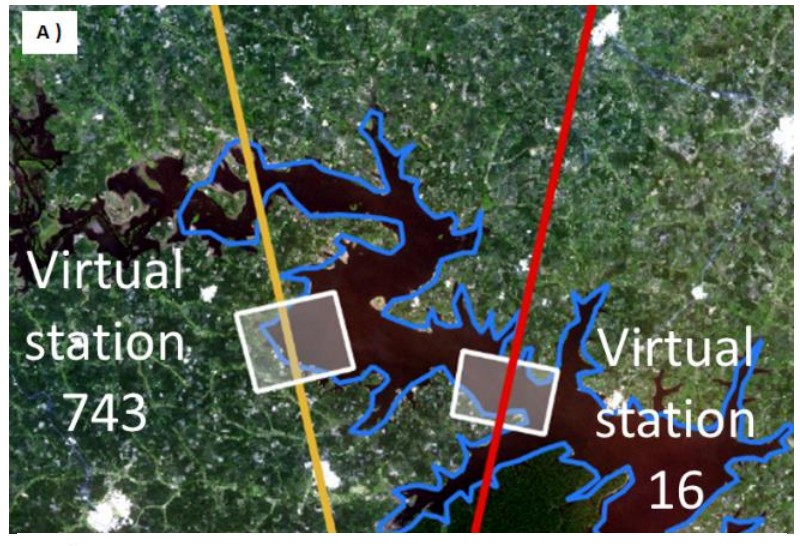

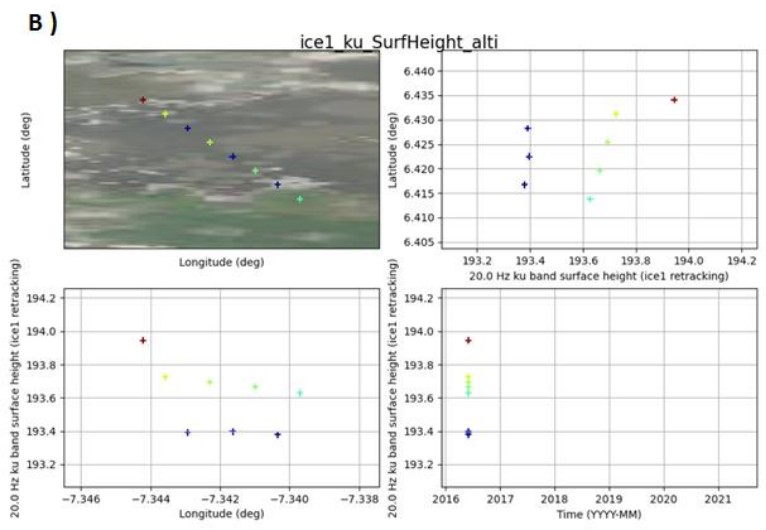

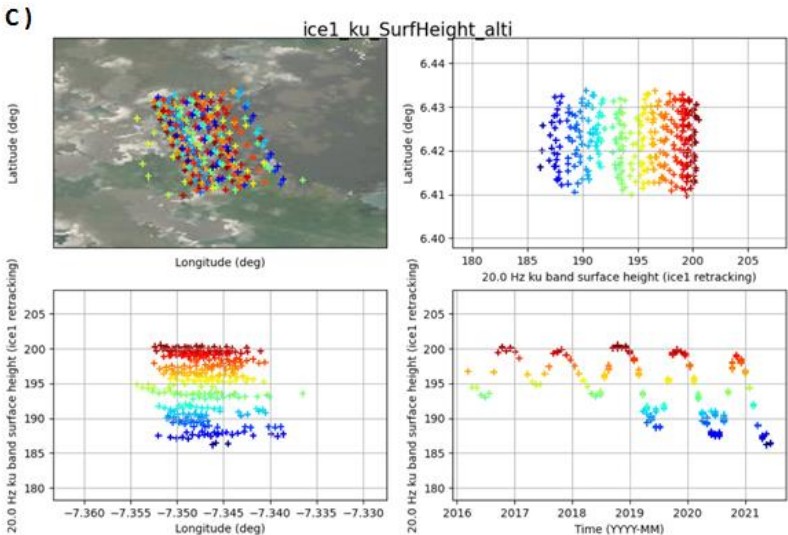

**Figure 3.** *Cont.*

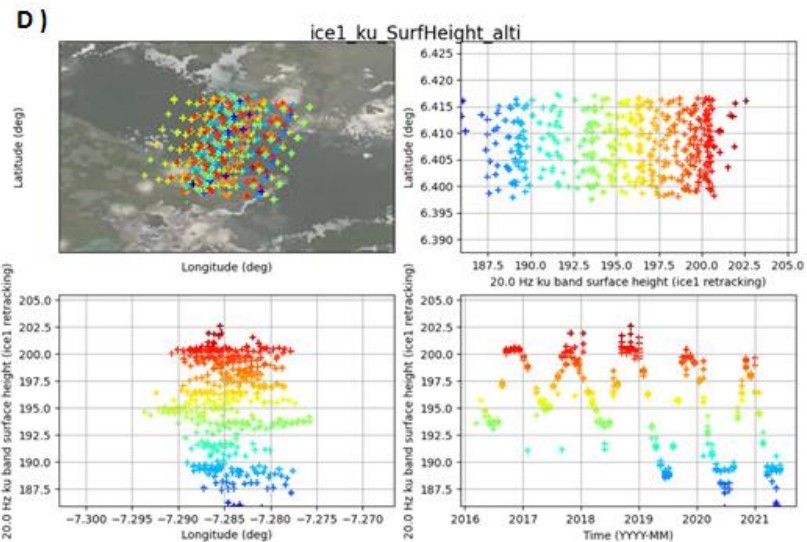

**Figure 3.** Water level time series extraction with Altis: (**A**) Delineation of the virtual station, (**B**) verification of valid points, (**C**) computation of water level time series with track 743 (orbit 372), (**D**) computation of water level time series with track 16 (orbit 8).

### 3.3. Altimetry Data Validation

For the validation, the data estimated by satellites and the in situ data collected at the DPE were first pre-processed (quality control and gap filling), then adjusted over the same period (16 June 2021) and at a monthly time interval. The validation of the data from altimetry processing was done in two steps: (i) a visual validation; (ii) a quantitative validation.

#### 3.3.1. Visual Validation

Visual validation consisted of making a visual comparison at monthly, seasonal and annual scales of the data estimated by radar altimetry with the in situ data. This step is necessary to perform a preliminary visual inspection of data trends over long time scales.

#### 3.3.2. Quantitative Validation

The quantitative validation used was a pairwise comparison that evaluated the performance of the altimetry products in estimating the lake levels. For this validation, as summarized in Table 1, the Pearson correlation coefficient $R^2$ was used to evaluate whether the radar altimetry estimates matched the in situ values. The bias and (RMSE) values indicated whether the altimeter product overestimated or underestimated the water levels on average. The NSE coefficient [42] was used to determine the percentage of variance between the altimeter product estimates and the DPE in situ observations.

**Table 1.** Quantitative validation criteria for water levels from altimeter measurements. $WL_{altimetry}$ = Water Level estimated by radar altimetry, $WL_{in\ situ}$ = Water Level recorded at the observation station, n = number of observations.

| Name | Formula | Validation Criteria |
|------|---------|---------------------|
| Bias | $BIAS = \frac{\sum_{i=1}^{n}\left(WL_{altimetry} - WL_{in\ situ}\right)}{n}$ | must tend to 0 |
| Root Mean Square Error | $RMSE = \sqrt{\frac{\sum_{i=1}^{n}\left(WL_{altimetry} - WL_{in\ situ}\right)^2}{n}}$ | must tend to 0 |
| Pearson correlation coefficient | $R^2 = \frac{\left[\sum_{i=1}^{n}\left(WL_{altimetry} - \overline{WL_{altimetry}}\right)\cdot\left(WL_{in\ situ} - \overline{WL_{in\ situ}}\right)\right]^2}{\sum_{i=1}^{n}\left(WL_{altimetry} - \overline{WL_{altimetry}}\right)^2 \cdot \sum_{i=1}^{n}\left(WL_{in\ situ} - \overline{WL_{in\ situ}}\right)^2}$ | must tend to 1 |
| NSE coefficient | $NASH = 1 - \frac{\sum_{i=1}^{n}\left(WL_{altimetry} - WL_{in\ situ}\right)^2}{\sum_{i=1}^{n}\left(WL_{in\ situ} - \overline{WL_{in\ situ}}\right)^2}$ | must tend to 1 |

## 4. Results

### 4.1. Agreement of Sentinel-3A Data with In Situ Measurements

The correlation coefficient $R^2$ between the in situ water levels and altimetry water levels given by the track 743 (orbit 372) was 0.99, and it was 0.98 for the track 16 (orbit 8) (Figure 4). The NSE coefficient was also equal to 0.99 and 0.98 for the tracks 743 and 16, respectively (orbit 8) (Table 2). For the bias and RMSE, the values obtained for tracks 743 and 16 varied from 0.13 m to 0.12 m and from 0.37 m to 0.67 m, respectively (Table 2).

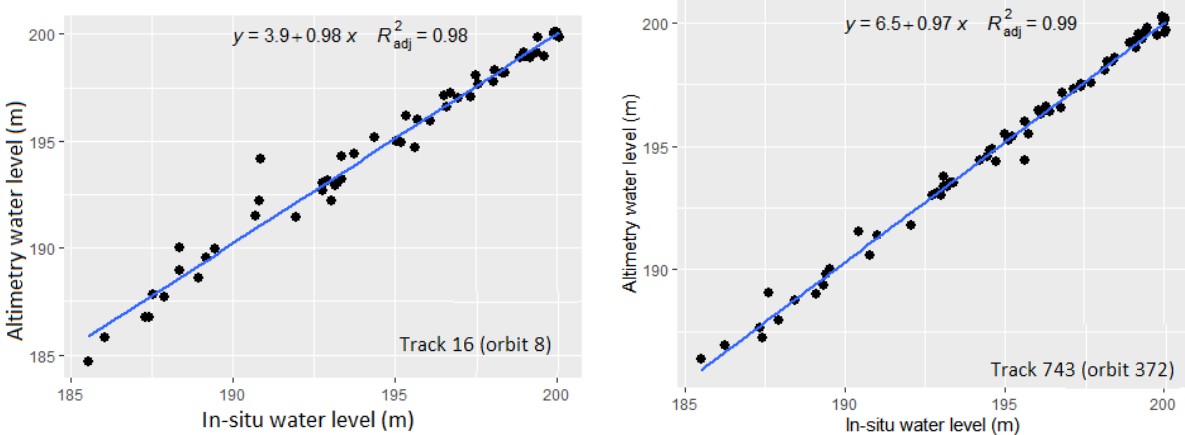

**Figure 4.** Concordance of Sentinel-3A altimetry water levels (tracks 16 (orbit 8)) and 743 (orbit 372)) with in situ water levels.

**Table 2.** Quantitative values of statistical parameters.

| Parameters | Track 16 (Orbit 8) | Track 743 (Orbit 372) |
|---|---|---|
| Bias | 0.12 m | 0.13 m |
| Root Mean Square Error (RMSE) | 0.67 m | 0.38 m |
| Pearson correlation coefficient ($R^2$) | 0.98 | 0.99 |
| NSE coefficient | 0.98 | 0.99 |

### 4.2. Evolution of Water Levels in Lake Buyo from 2016 to 2021

The variations in the water levels of Lake Buyo over the period from 2016 to 2021 are presented in Figure 5. Analysis of this figure indicates that from 2016 to 2020, the maximum water levels in the lake varied only a little; they were around 200 m. Meanwhile, the minimum water levels varied greatly; they varied from 192.95 m (2017) to 188.35 m (2019). From 2020, we observed a decrease in the maximum levels, dropping from 200 m to 198.03 m. Regarding the minimum heights, they continued their visible decrease from 2018, and went from 188.35 m (2019) to less than 185 m (in the first half of 2021). Considering the maximum and minimum values that were reached, which went from 200 m to 198.03 and from 192.95 m to less than 185 mm, respectively, a decrease in the water levels of Lake Buyo over the period from 2016 to 2021 was observed. However, given the limited time interval (about 5 years), it is premature to make a conclusion on the general trend of the water level evolution of the lake.

### 4.3. Annual and Seasonal Variations in Water Levels of Lake Buyo

The altimetry water level recordings of Lake Buyo from Sentinel-3 started on June 02, 2016, which was a few months after the launch of Sentinel-3A; this explains the absence of measurements from the first five months of the year (Figure 6a). Water levels during the second half of the year gradually increased from July (192.78 m) to reach their maximum level in November (200.06 m) (Figure 6a). This phase of increase in the water levels of the lake occurred just after the long rainy season (July). From December, the decrease phase of the water levels in the lake began. This period corresponded to the beginning of the long dry season, which is generally between November and December.

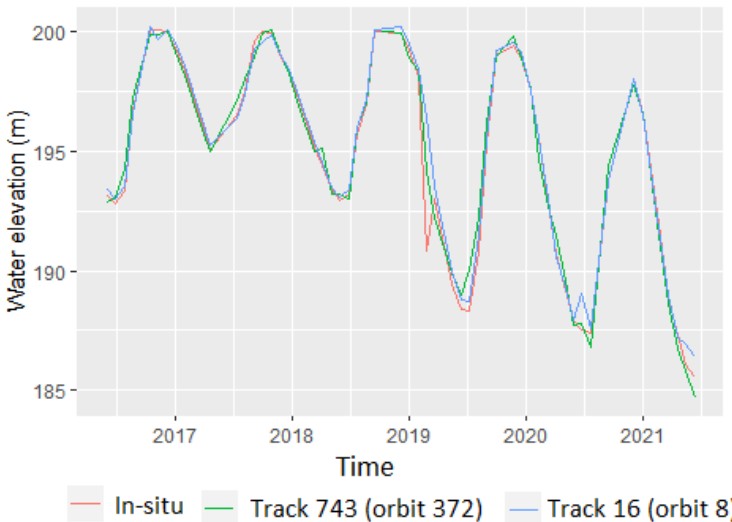

**Figure 5.** Superposition of the altimetry water levels from tracks 16 (orbit 8) and 743 (orbit 372) with the in situ water levels.

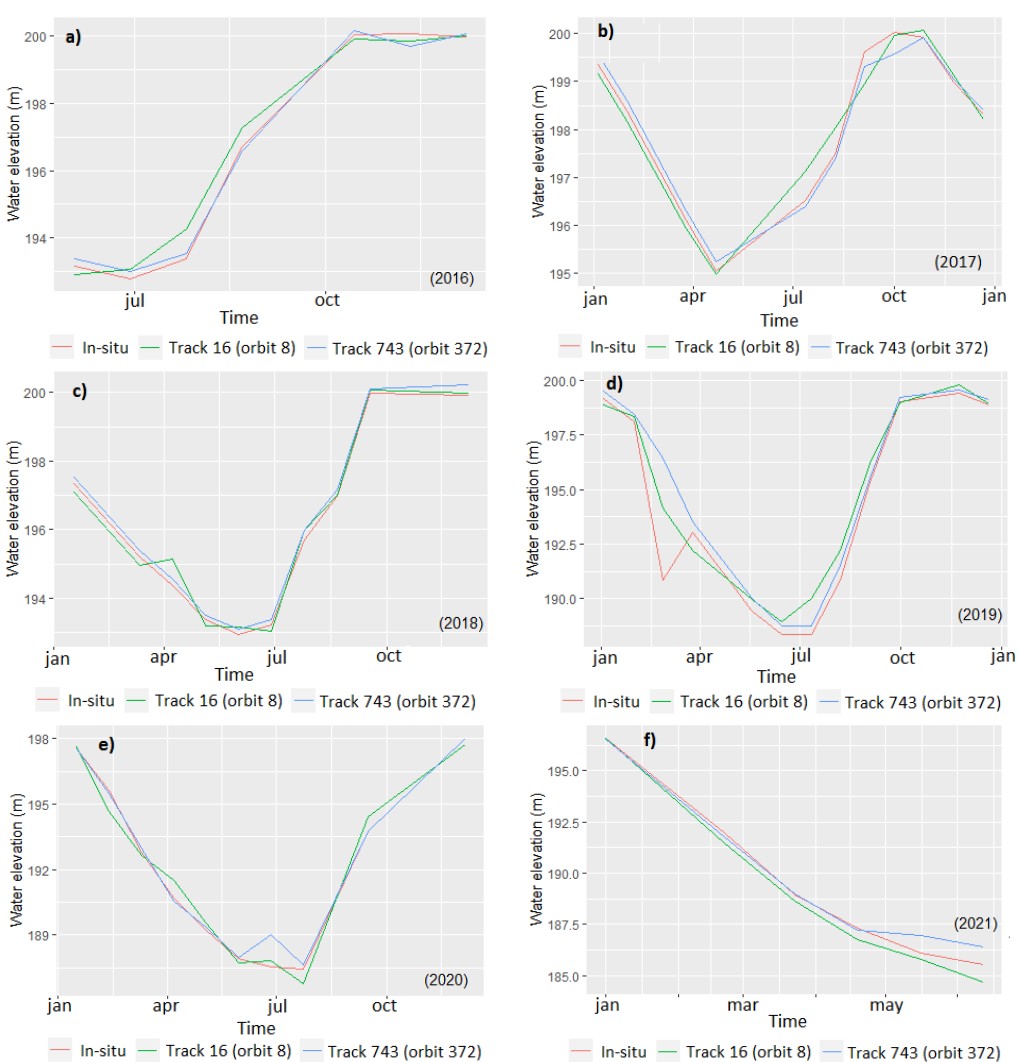

**Figure 6.** Annual variations in water levels in the lake in: (**a**) 2016; (**b**) 2017; (**c**) 2018; (**d**) 2019; (**e**) 2020; (**f**) 2021.

At the beginning of 2017, the decrease in the water levels of the lake that started in December 2016 continued gradually until April 2017, when the minimum level of the lake was reached (195.04 m) (Figure 6b). After this decrease, an increase in the water levels started from May and continued gradually until October, when the maximum value was reached (200.02 m) (Figure 6b). In contrast with 2016, the beginning of the increase in the lake's water levels coincided, in 2017, with the beginning of the long rainy season (April) and not the end (July). The same was true for the beginning of the decrease in the water levels, which also coincided with the beginning of the long dry season, which most often occurs between November and December (Figure 6b). The maximum water level recorded in 2017 (200.02 m) was almost identical to that of 2016 (200.06 m), while the minimum water level in 2017 (195.04) was higher than that in 2016 (192.78 m).

The temporal variations in the lake's water levels in 2018, 2019 and 2020 also evolved according to the two main seasons of the year: the long rainy season from April to July, and the long dry season from November to March. In fact, the decreases in the water levels started most often between November and December (Figure 6c–e). These decreases then continued regularly until June–July of the following year, when the lake reached its minimum level. This level was 192, 95 m, 188.35 m and 187.42 for the years 2018, 2019 and 2020, respectively. These recession phases were immediately followed by water level increases, which occurred most often between July and August. These increases took place steadily, with the lake reaching its maximum level between October and November. This level was approximately 200 m in 2018 and 2019, and 198.03 m in 2020 (Figure 6c,d,f).

The temporal variations in the lake's water levels in 2021, for which the available altimetry data only covered the first half of the year, seemed to evolve in the same way as in previous years. In fact, as seen in Figure 6e, a decrease in the water levels that seemed to continue until July was observed. This recession started in December 2020 (Figure 6).

In addition, over all the years covered by this study, the short dry season, which most often extends from July to September, did not seem to influence the general trend of the lake level curve. The same was observed for the short rainy season, which runs between September and October. During these two short seasons (dry and rainy) the water levels in the lake continued to increase in a regular and constant way to reach their maximum level in October or November, in spite of the low amount of precipitation recorded (Figure 6). This may be explained by the amount of water brought to the lake by the Sassandra River after the main rainy season. This quantity of water, which continues to feed the lake after the long rainy season, is greater than the inflow due to rainfall during the short dry and rainy seasons. The influence of the Sassandra River on Lake Buyo was also visible at the beginning of the long rainy season. Indeed, despite the beginning of the long rainy season in March or April, the water levels in the lake continued to decline until June–July, which almost corresponds to the end of the long rainy season. The lake thus reached its maximum level in October for almost all the years studied (Figure 6). The time lag between the end of the long rainy season (July), when the Sassandra River reaches its maximum level and the lake records its maximum level (October), is about four months.

Figure 7 provides a summary of the changes in Lake Buyo's water levels from 2016 to 2021. The general appearance of their water level box plots was almost identical for tracks 16 (orbit 8) and 743 (orbit 372) and the in situ measurements. From October to January, the lake's water levels changed very little, with a median between 200 m and 198 m obtained, representing a fluctuation of 2 m. From January to June, the decreases in the water levels became very fast, with a median between 198 m and 188 m recorded, which represents a fluctuation of 10 m. The increases in the water level from June to October was as fast as the decreases. The median varied between 188 m and 200 m, which represents a fluctuation of 12 m. The filling and draining of Lake Buyo occurred at roughly the same rate, and were controlled by the two main tributaries of the Sassandra watershed (N'zo and Sassandra). The seasonal variations in rainfall in the region of the lake seemed to have limited influence on the variations in the water levels of the Lake.

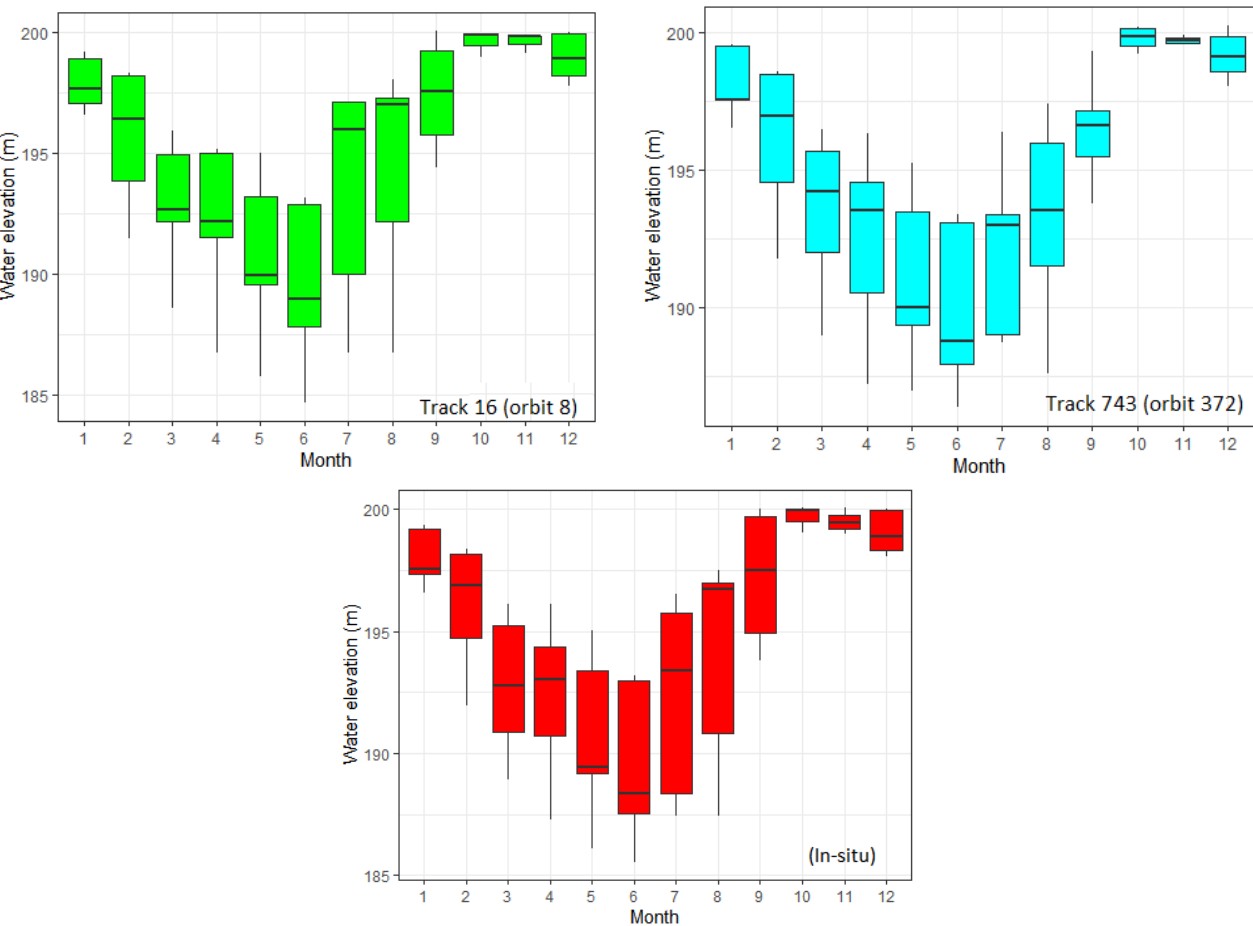

**Figure 7.** Monthly variations in the lake's water levels from 2016 to 2021 (with track 16 (orbit 8), track 743 (orbit 372) and in situ measurements).

## 5. Discussion

The altimetry water levels estimated by Sentinel-3A were in good agreement with those measured in situ. Indeed, the $R^2$ correlation coefficients and the NSE coefficient oscillated between 0.84 and 0.98 and between 0.80 and 0.98, respectively. The bias and RMSE values varied from 0.12 m to 0.75 m and from 0.67 m to 1.95 m. respectively. These results are similar to those obtained by [40] for the Ogooué watershed in Gabon (Central Africa), who obtained $R^2$ correlation coefficients ranging from 0.88 to 0.98 and RMSE values oscillating between 0.20 m and 0.41 m using Sentinel-3A altimeter data. The same was observed by [41], whose results showed a very good agreement between the in situ and altimetry water levels from Sentinel-3A in the Inner Niger Delta; they obtained $R^2$ correlation coefficients greater than 0.9 and RMSE values less than 0.7 m in 88% of cases. The authors of [27] obtained a good estimate of the water levels in a coastal lagoon located in southeastern France using CryoSat-2 altimetry data, recording an $R^2$ correlation coefficient estimated at 0.79 and an RMSE value less than 0.44 m. Water levels obtained by [40] for the Ogooué watershed, using CryoSat-2 altimetry data, were even better than those calculated by [27], recording an $R^2$ correlation coefficient equal to 0.96 and an RMSE value less than 0.26 m. The authors of [39] also achieved a good performance in estimating water levels in lakes located in mountainous areas using Sentinel-3A/3B altimetry data, recording $R^2$ correlation coefficients that were most often greater than 0.85 and RMSE values generally less than 0.07 m. From a technological point of view, Cryosat and Sentinel-3 both use the Delay Doppler processing technique. However, Sentinel-3 has two major advantages over Cryosat-2. First, its satellite resource (memories, etc.) have been sized so that data can be acquired/recorded all along the orbit. In addition, Sentinel-3 has a

tracking mode (called open loop mode), which is essential for consistently capturing data over hydrological targets.

The ALTiS software used in this work is a simple but efficient piece of software (http://ctoh.legos.obs-mip.fr/applications/land_surfaces/softwares/altis). ALTiS uses level-2 GDR data that is freely available. It displays the variables contained in this data, including H, $R_0$, and the various corrections applied to $R_0$ and h calculated automatically, as well as several other variables, such as backscatter coefficients, brightness temperatures at the various frequencies measured by the radiometer on board the satellite platform, etc. [39]. In addition, it generates water level time series by calculating the median and mean values of the points inside the virtual stations for each cycle. In this study, the time series of water levels was computed using median values rather than mean values. Many studies indicate that the best results are obtained with median values rather than mean values [24,27,40].

The Sentinel-3A satellite, which has a repeatability of 27 days, allowed this study to highlight on the one hand the seasonal variability of Lake Buyo's water levels, and on the other hand the influence of the Sassandra River on the lake. This temporal resolution proved to be sufficient to highlight the influence of the different seasons on the water levels of Lake Buyo, for which the duration of the seasons varies from two to five months at most. Sentinel-3 altimetry data have also been used to identify seasonal and interannual variability in the water levels of the Niger Delta [41] and in Central Africa [40], where seasons can last longer than six months.

## 6. Conclusions

In this study, the seasonal and interannual variability of water levels in Lake Buyo was studied using Sentinel-3A radar altimetry data. The extraction of the altimetry water level time series and the corrections applied to these altimetry data (geophysical and environmental corrections) were performed with the ALTiS software. The validity of these data was evaluated by calculating several statistical parameters, such as the $R^2$ correlation coefficient, the NSE coefficient, the RMSE and the bias. The values obtained indicated a good overall agreement between the Sentinel-3A altimetry water levels and the in situ water levels, with $R^2$ values between 0.98 and 0.99 obtained. Further, the NSE coefficient also ranged from 0.98 to 0.99, the RMSE values ranged between 0.38 m and 0.67 m, and the bias values oscillated between 0.12 m and 0.13 m. Two important periods were identified during this study, namely March–October and November–March. The first period from March to October corresponded to an increase in the water levels of the lake. The maximum water level was generally reached in November after the last rain of October. This maximum value varied from 200.06 m (2016) to 198.03 m (2020). The second period, from November to March, was the phase that saw a decrease in the lake's water levels. This decline most often started between November and December, and then continued steadily until June–July of the following year, when the lake reached its minimum level. The minimum water level ranged from 195.04 m (2017) to 187.42 m (2020). The influence of the Sassandra River on Lake Buyo was visible, especially at the beginning of the long rainy season. Despite the start of the long rainy season in March, the water levels of the lake continued to decrease until July, which almost corresponds to the end of the long rainy season. The time gap between the end of the long rainy season (July), when the Sassandra River reached its maximum level, and the lake reaching its maximum level (October), was about four months.

Radar altimetry data, and more particularly Sentinel-3A data, proved to be efficient during this study. The results show that radar altimetry can be used to study water level variations in surface hydrological systems (lakes, rivers). The technological advances of the latest altimetry missions (SARAL, Sentinel-3A/3B, Cryosat-2) and the performance of current altimetry data processing algorithms (BRAT, GPD plus, ALTiS) now make it possible to obtain results that are increasingly close to reality. Radar altimetry today, therefore, provides an undeniable opportunity for the monitoring of water level variations

in oceans and continental waters (lakes, rivers), as it allows for significant savings compared to the installation and maintenance costs of traditional gauging stations.

**Author Contributions:** S.O., V.-C.J.S. and K.F.K. conceived of and designed the study. S.O., T.C., B.M. and B.P. designed and developed the methodology. S.O., C.A.K.K. and V.-C.J.S. performed the altimetry data processing. All authors analyzed the results and contributed to the writing of the paper. All authors have read and agreed to the published version of the manuscript.

**Funding:** This research was funded by SISEB-CI (Suivi par Imagerie Satellitaire optique et radar du niveau et du volume d'Eau des Barrages de Buyo et Kossou en Côte d'Ivoire).

**Data Availability Statement:** All data supporting the figures and graphics are available via the cited references.

**Acknowledgments:** The authors are grateful to the Center for Topographic studies of the Ocean and Hydrosphere (CTOH) at LEGOS (Toulouse, France) for providing the altimetry dataset. We also thank the Direction de la Production de l'Electricité de Côte d'Ivoire for supplying the Lake Buyo water levels.

**Conflicts of Interest:** The authors declare no conflict of interest.

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
