# Peer review of "Contribution of Sentinel-3A Radar Altimetry Data to the Study of the Water Level Variations in Lake Buyo (West of Côte d’Ivoire)"

_remotesensing, doi:10.3390/rs14215602_

Round 1

Reviewer 2 Report

It is an interesting work focusing on a specific application of satellite radar altimetry in a region characterized by a growing interest in this technique, thus with a good potential interest to the readers.

 The overall perception is good enough; but the work still needs some improvements, being essentially a case study and not carrying advancements or further investigation in the technique.

In particular, I feel that an improvement in the description of the method and the processing of the data used in this paper is needed, and would be much appreciated by the readers.

Thus I’m recommending major revisions.

 My specific comments follow. You will see that some are very minor, others instead represent key aspects of the paper.

 At line 74 please replace “It covers…” with “Satellite radar altimetry covers…”

-        Both Figures 1 and 2 are too much affected by clouds and hardly understandable by the readers. Moreover, the water body contours by Google are very different from the observable contours in the image. I understand that it’s hard to find good and cloud-less mosaics in the historical images of Google Earth in your area. I suggest the authors to look for good Sentinel2 or Landsat8/9 passages and produce their own RGB images, also contouring the water body properly.

 I attach an example of a less-cloudy picture reporting the two S3A orbits of your interest, where you can still observe the uncorrect water contouring mentioned above.

 At lines 137-138: The sentence “These measurements… (Figure 2)” is rather generic. Here authors should specify the posting rate of the data used (the extension ‘ku_20’ in Figure 3 makes think that’s perhaps 20Hz). These fields can also have a posting rate 80Hz in the GDR. Please check and specify in the text. Given the posting rate, I suspect that you have multiple valid measurement points per each passage and your software averages them (by median filtering?). This thing and your strategy to select valid and reject invalid points (e.g. near-coast points in ascending/descending orbits) should be well clarified in this section.

-        Also, connected with the comment above: did you check or analyse radargrams to identify invalid radar returns?

-        Lines 140 and 143: perhaps when you say ‘altimeter’ you mean ‘hydrometric’ measurements.

-        At line 189 please say: “The main geophysical corrections…” They’re actually not the only ones.

-        At line 215, when you say “cross-section” maybe you mean “crossing”

-        Line 251: “It consisted…” is not a good way to begin a section. Please rephrase specifying the subject of your sentence.

-        Line 253: this sentence makes the whole sub-section 3.3.1 unclear. If you do correlation (that is a quantitative calculation) what’s the sense of a visual inspection of the data? Maybe you mean that you’re doing a preliminary visual inspection of the data trends at long time scale?

-        Line 261: please instead of just mentioning “The Nash” specify that you’re dealing with the Nash-Sutcliffe Efficiency coefficient (NSE), and try to add at least a reference about it.

-        As a general idea, I invite the authors to speak in terms of “relative orbit” numbers, instead of “pass numbers” and call them “tracks”. This may be misleading to many readers. May I suggest to identify the two orbits that you selected in terms of “relative orbit number 8” and “relative orbit number 372”.

-        Lines 273-283: I agree on choosing track 16 (rel. orbit 8), but the proximity to the in situ gauge does not look like a convincing reason. Data related to orbit 372 (Figure 5) seem more spread (higher variance?) compared to data related to orbit 8. This may be the hidden reason of your discrepancy and should be somehow discussed in the paper. Looks like some issue with the preliminary selection of data points. Please see also the comment related to lines 137-138.

-        Line 323: “…given the narrowness of the study interval…” maybe you mean “…given the limited time interval…”

-        Lines 368-369: “…lake level in 2021, for which the available altimetry data only cover the first half of the year…” Since this paper is potentially published in late 2022, I suggest to extend this dataset to late 2021, acquiring the additional data.

-        Line 380: “…during the short dry and rainy seasons” Please remove the ‘s’ from ‘shorts’.

-        Figure 8: I suggest to place an inset in each sub-figure specifying the corresponding year of each plot. Currently there are some x-axis labels containing the year, some others do not.

-        There also few typos or some minor grammar issues, such as in line 74 “Sar Radar Altimete”, or many missing commas before the word “which”. I recommend to check through all the text.

As to the references, some expansion is strongly suggested. In particular, some of the cited works like 1-8, 12 and 14-15 represent correctly a general coverage of basic aspects and studies, but part of the authors are repeated and belong to a very narrow community. I suggest to keep them and add other sources from the broader altimetry community.

Also, by completing the reference list you may consider more recent and adequate references to add to 7 and 12 (that is a chapter from ref. 7), because both are certainly fine, but rather old, while good reference literature has been produced in these last 20 years about altimetry for inland and coastal applications.

Reviewer 3 Report

The main point is really to understand what is going on with track 743 since it was correct at the beginning and similar issues might occur with track 16. In addition, the influence of the dam (if any) is not clearly detailed so at the end it is not known if all effects are linked to rain falls/Sassandra river.

Round 2

Reviewer 1 Report

Comment #1: Please make sure the x-labels in updated Figure 6 are consistent. For some subplots, the year number is present whereas for some it is not present. Make it consistent for all subplots. Also, keep the x-labels in English. 

Comment #2: For updated Figure 7, the caption does not match with the figure contents. Please correct it.

Comment #3: Line 470-488 - Please add a few sentences explaining the boxplots from Figure 7 and the subtle differences in Track 16, Track 743 and in-situ. 

Reviewer 2 Report

I appreciate the efforts by the authors to follow the comments made.

But, I still see two important comments unanswered and no actions taken. In particular, about the usage of pass numbers and calling them “tracks”, and about expanding the references to a broader altimetry community. Readers may take benefit if you consider recent generic literature about altimetry for inland and coastal applications, going further a mere list of case studies. I.e., here I mostly see case studies in France, US, some African countries and Switzerland, while good reference books, book chapters and review works with more general coverage are missing. Instead, I find appropriate the references concerning the basics of radar altimetry, SAR altimetry and corrections.

Thus, I’m asking to provide a reply before acceptance of this paper. If the authors do not think to apply the  suggestions, at least should give a reply with an explanation.

Reviewer 3 Report

It would great to get a clean copy without any red marking to make a final reading. Overall it is really excellent that the 2 tracks are now in good agreement. We can observe that the RMSE errors are a bit on the high side but there is nothing, which can be done on this since this is a product from ALTiS. Hopefully, at some point, better quality products will be provided to users. Just a few comments on the revised copy:

a)line 35: we are using Stucliffe Efficiency coefficient but on line 317 it the Nash Sutcliffe....better to use define it once and then use the acronym elsewhere

b)line 66,67, 112 (maybe other places), what do we mean by square root km (racine carrée?) before it was km2...this unit makes much more sense for a surface

c)page 9: the figures should be re-ordered or put in an annex since it looks visually messy. Some abscissas are displaying Time (YYYY-MM)...this does not look good....just write Time. Sometimes we are using upper case A) B), D) and sometimes lower case c) is used

d)page 15: all my apologies for being painful but the abscissas are not consistent sometimes it is month only or sometimes it is month/Year. It would be great to be consistent

e)pages 17/18: Figure 7 seems to be linked to the plot on page 17. We are missing a title for the plot on page 18. I am also referring where in the article this plot is referred to?

As stated above, I would really need to have a copy with no marking to make a proper review.
